# Reassessing and Extending the Composite Rainfall Record of Manchester, Northwest England: 1786–Present

Neil Macdonald [1],* and Robert Dietz [1,2]

1 Department of Geography and Planning, School of Environmental Sciences, University of Liverpool, Liverpool L69 7ZT, UK
2 AtkinsRéalis, Birchwood Park, Chadwick House, Warrington Rd, Risley, Warrington WA3 6AE, UK
* Correspondence: neil.macdonald@liverpool.ac.uk

**Abstract:** A monthly composite rainfall record for the period 1786–present representative of Manchester, northwest England is presented. The 235-year record ranks as the second-longest instrumental rainfall record available for northern England, and the fifth-longest for the UK, and contributes to a growing network of long homogenous rainfall series. A composite record is constructed, extended, and homogenised, and the record is analysed in terms of annual and seasonal variability, with a focus on extreme wet/dry events. Three primary meteorological stations in Manchester, located within 2 km of one another, form the basis of the reconstruction, with other records identified for infilling and extension based on their longevity, continuity, and proximity to the primary stations. A linear regression analysis is applied to produce a continuous record, and adjustment factors are applied to ensure homogeneity. Record homogeneity is assessed via cross-comparison with long-term records from the region (Carlisle, Chatsworth House and HadNWEP), and the methods are applied to assess relative homogeneity include the double-mass curve and Standard Normal Homogeneity tests. The Manchester record is deemed to be homogenous overall but includes two periods of increased uncertainty: 1786–1819, comprising the earliest observations and greatest number of different stations, and 1883–1911, which encompasses multi-year and multi-decadal drought events of (1883–1885 and 1890–1910) as identified by other long-term meteorological studies. The analysis of the entire record reflects long-term rainfall variability with an increasing, although not significant, trend in annual rainfall observed. Seasonally, a significant increase in winter rainfall is exhibited, in keeping with patterns observed in other regional studies. Seasonal rainfall totals are found to be highly variable at the decadal timescale. Several well-documented extreme wet (e.g., autumn 2000) and dry (e.g., summer 1976) seasons are identified, including historic events (e.g., the floods of summer 1872 and drought of summer 1887) from the less-well documented eighteenth and nineteenth centuries.

**Keywords:** Manchester; reconstruction; precipitation; climate; long-series; rainfall

## 1. Introduction

In recent decades interest in past climate variability and climate change has grown as a consequence of global warming linked to anthropogenic activity [1]. Studies have used historical documentary evidence and instrumental records to gain invaluable insight into how climatic parameters, for example temperature and precipitation, have changed through time, thus providing a better appreciation of climatic variability [2,3]. Through the analysis of long-term global climate records, a greater understanding of the anthropogenic contribution to climate change can be ascertained [1]. On a small scale, individual records from single meteorological stations are important for studies of local or regional records of change [4]. Furthermore, the incorporation of these records into larger networks provides us with a greater understanding of climate variation on national and global scales [5,6], of climate forcing, and the impact of natural and anthropogenic factors [7].

Manchester is a major city within the metropolitan borough of Greater Manchester in northwest England. It has a rich history of weather recording, reflecting the influence of rainfall in the development of the city through agriculture, early industrialisation, subsequent urban expansion, and potable water supply needs, which have contributed to the prosperity of the city. A composite monthly rainfall series for Manchester (1765–1971) was originally constructed by the eminent climatologist Gordon Manley [8], published in the format of monthly totals, expressed as percentages of the years' rainfall to the nearest whole number, from 1765 to 1971. Manley [8] provided a detailed description of how the composite series was constructed, including dates, names of recorders, locations, and estimated averages for locations with respect to 1916–1950, and often included metadata regarding the gauge where applicable. However, great uncertainty exists within the record, regarding the reliability and homogeneity of the series, especially for the early part of the record. The record was not constructed using modern homogenisation techniques nor has it undergone homogenisation testing (e.g., double-mass curve tests). To increase the reliability and accuracy of the Manchester rainfall record, revision and extension have been undertaken using homogenisation methods [3]. The revision and extension of the Manchester rainfall record is important (i) for the understanding of the long-term temporal variability of rainfall in Manchester; (ii) for the understanding of annual and seasonal variations informing water resource management decisions; and (iii) because the full series is approximately 75 years longer than most climate series used for analysing long-term climatic variability in the UK (the majority starting in 1860) and therefore can be used to improve the understanding of natural variations in the climate of the region; of particular interest is the poorly documented severe drought of the 1780s and 1790s [9]. The principal aim is to produce a homogenous, continuous composite monthly rainfall series for Manchester 1786 to present. This was achieved by:

1. Revisiting the original sources of historic rainfall data identified by Manley [8], coupled with the acquisition of contemporary meteorological station data for Manchester.
2. Reconstructing the record using statistical techniques to infill and bridge gaps between station records.
3. Statistically assessing the homogeneity of the record through comparisons with other long-term homogenous rainfall series representative of the region; Carlisle 1757–2012 from Todd et al. [3], and Chatsworth House, 1777–2015 [9], and the Hadley Centre's northwest England rainfall series (HadNWEP 1873–2015 from Alexander & Jones [10] and Simpson & Jones [11]).
4. Statistically analysing the series to determine long-term seasonal and annual variability, focusing on extreme wet/dry seasons and events, and to identify long-term trends.

## 2. Historical Instrumental Rainfall Observations and Reconstructions

The practice of recording the weather can be traced back as far as the 17th century in Europe following the invention of reliable measuring instruments, notably the rain gauge by Father Benedetto Castelli, a pupil of Galileo, in 1639 [12]. The use of such instruments was initially limited by the Inquisition after Galileo's trial and subsequent sentencing for heresy in 1633 [12]. The *Medici Network* was the first international meteorological network composed of measurements from eleven stations across central Europe (1654–1667), with measurements collected using a system devised by the scientific academy (*Accademia del Cimento*) founded by the brothers Ferdinand II (Grand Duke of Tuscany) and his brother Prince Leopold de' Medici. Following the closure of the *Accademia del Cimento* by the Inquisition in 1667, much scientific activity ceased, including meteorological observations by the *Medici Network* [12,13]. Consequently, it was not until the early 18th century that regular instrumental rainfall observations were once again being undertaken across Europe, with at least ten sites recording in the UK by 1720 [14]. The earliest record for the UK was from Burnley in northwest England, where Richard Towneley kept precipitation records between 1677–1704 [15]. Recent studies have been able to use even the earliest rainfall measurements to reconstruct long, continuous records across Europe. One notable example

being Camuffo et al.'s [12] 300-year series representative of the West Mediterranean basin, which incorporated fourteen long series from six different subareas. However, the practice of producing rainfall records that incorporate early measurements, each of which is likely to contain unique sources of error that may change through time, requires care and consideration [16]. Craddock and Wales-Smith [17] state that there are approximately twenty potential sources of error at any rainfall station, ranging from defects in the construction to changes in exposure, and it is almost impossible to decipher what may have resulted in a deficient catch at historic sites without visible inspection and/or a careful analysis of the metadata. Metadata may include details regarding changes in instrumentation, location, observer, or surroundings [18]. Few sites exist where early observations continued for extended lengths of time; thus, composite rainfall series rely on the presence of records in nearby areas for the construction of a single homogenous precipitation series that can include even the earliest observations [3].

The British Rainfall Organisation founded by George Symons in 1860 (amalgamated into the Meteorological Office in 1919) served as a representative body for early observers and published annual volumes under various titles (originally Symon's 'British Rainfall', eventually shortened to 'Rainfall') from 1861 to 1991 [19]. The majority of long records were constructed during the 1970s, including single station studies, for example, Oxford's Radcliffe Observatory (1767–1814, [20] and composite series, like that of Manchester (1765–1971, [8]), with new and old series being revisited in recent years, such as Kew [21,22], Oxford [23], Carlisle [3], Durham [23], Chatsworth House [9] and Stornoway [24]. Several long rainfall records exist for northwest England, notably the Carlisle, 1757- [3] and for Chatsworth House, 1777- [9], the 200-year index series for the English Lake District [25], and the series representative of the region developed by the Hadley Centre (HadNWEP, [10]). There are also opportunities for several additional sites in the UK to be revisited, with opportunities to extend rainfall series back to the eighteenth century for cities such as Liverpool and Edinburgh.

Appreciating the value of the records, and of national rainfall variability, Craddock [14] grouped reliable monthly rainfall records from twenty sites across eleven UK regions into the first catalogue of long-term precipitation series; the compilation, revision, and the extension of such records has continued to the present day [11,26–28], highlighting the usefulness of such records for examining the relationship between changes in rainfall intensity and extreme events with regard to informing the management of flood risks and water resources [27,29].

### 3. Methodology

The majority of the data were collected from the UK Meteorological Office archives (Devon Heritage Centre), Exeter, and the Manchester University John Rylands Library archives, Manchester. Datasets were collected in the form of meteorological observations and the associated metadata from a number of sites across Manchester. Metadata in this case are defined as station documentation containing information about the data, for example, how, where, when, and by whom it was collected [18]. In most cases, sufficiently detailed metadata were provided by Manley's [8] notes concerning changes in the rain gauge, recording practices, or location; however, supplementary information was yielded from documentary sources, such as the Manchester Medical Collection biographies and the early Transactions of the Manchester Philosophical Society.

The meteorological data were collected from several sources, ranging from archival to published and digital sources. Manley's [8] list of available material, including dates, locations, and observers, was useful in refining the search. The bulk of the rainfall data (1765–1957) was available in the form of archived copies at the UK Meteorological Office, with data photographed from the 10-year rainfall books and later transcribed and digitalised, as well as any accompanying metadata (Table 1). In addition, some of Dr. John Dalton's (1819) original measurements (1794–1818) were available at the UK Meteorological Office, having been published in the Manchester Literary and Philosoph-

ical Society's journal, as well as some of the early Whitworth Park data (1893–1904 in monthly form, and 1910–1936 in daily form), which was archived at the John Rylands Library (Figure 1). Data for the latter part of the record (1958–2015) were extracted from the British Atmospheric Data Centre's (BADC) Met Office Integrated Data Archive System (MIDAS) Land and Marine Surface Station online digital repository (UK Meteorological Office). For the purposes of testing the homogeneity of the Manchester record, other long-term homogenous rainfall records from the region were required. Todd et al.'s [3] Carlisle record (1757–2012) and the Hadley Centre's regional precipitation series for the northwest of England (HadNWEP: 1873–present) were also extracted. The majority of early meteorological measurements were recorded using imperial units of measurement, inches, with these converted to millimetres. The daily data were converted to a monthly resolution to keep the dataset consistent and limit potential issues surrounding the heterogeneity of daily rainfall patterns.

**Table 1.** All sources for rainfall data used to reconstruct the composite Manchester rainfall record. Primary sources highlighted in bold.

| Site (Station ID) | Observer | Years | Notes |
|---|---|---|---|
| Greengate, Salford | George Walker | 1786–1793 | Start of composite record. |
| New College, Dawson Street (also known as Mosley St) | John Dalton | 1794–1802 | Details unknown; "similar to Kendal gauge?" [8]. Kendal gauge: 10 in diameter. |
| **Mayfield** | **John Dalton** | **1803–1840** | **Missing data: Mar–Apr 1807, Dec 1809, and Jan–Feb 1811. Filled in using linear regression with Hanson's record at Lying-in hospital 1.86 km away (below). Rain gauge: 10 in diameter, on bench >2 ft off ground in Thomas Hoyle's garden, fairly open. Small changes in exposure through time [8].** |
| Lying-in Hospital | Thomas Henry Hanson | 1807–1811 | Used to infill Mayfield. Gauge: 6 in diameter, on hospital roof. |
| Market Street | Joseph Casartelli | 1841–1852 | Gauge: 5 in diameter, 3 ft above ground, sheltered by high wall to the east. |
| **Ardwick** | **Joseph Casartelli** | **1853–1879** | **Gauge: 8.5 in diameter, 3 ft above ground [8].** |
| Gorton Reservoir (11,971) | J. Bateman/ G.H. Hill | 1880–1892 | Gauge: 12 in diameter, 2 ft above ground. |
| **Whitworth Park Observatory (16,844)** | **Manchester University** | **1893–1957** | **Gauge: 8 in diameter, 1 ft above ground.** |
| Ringway Airport (1135) | Meteorological Office | 1951–2004 | Used to infill Denton gap. Gauge: 5 in diameter, 1ft above ground. |
| Denton Reservoir (11,958) | Manchester Waterworks | 1958–2023 | Missing data: 1960-Jan 1979. Filled using LR with Ringway 15.09 km SW. Gauge at 306 ft, below embankment previously level with, generally 2 ft above ground [8]. |

The stations were assessed based on their longevity, continuity, and proximity to one another, with monthly means with standard errors calculated. The results were analysed to assess which stations would be included in the reconstruction. The infilling and extending of the primary station records were conducted via linear regression. Linear regression is a statistical method that has been commonly applied in studies on long meteorological time series in the UK [3]. The linear regression model applied in this study is the same as that of Macdonald et al. [30] and Todd et al. [3]; a simple linear regression, assessing the relationship between a single dependent variable ($y$) and a single independent variable ($x$) to infill and bridge gaps in the primary station (dependent variable) record using data from a neighbouring overlapping station (independent variable). Using the longest possible overlap period, a scatter plot of mean monthly rainfall data with a linear trend line and regression equation was produced. When deciding whether the correlation value indicated a strong enough relationship between the primary and neighbouring station for the latter to be accepted for infilling and extending the primary series, Peterson and Easterling [31]

recommend that a threshold of greater than or equal to 0.8 be met for the neighbouring station to be deemed sufficiently reliable for use. In this study all correlation values exceeded this value, increasing confidence in the reconstruction (Table 2). Once a suitable neighbouring station was identified, the linear regression equation could be used to infill periods of missing data and bridge gaps between primary stations.

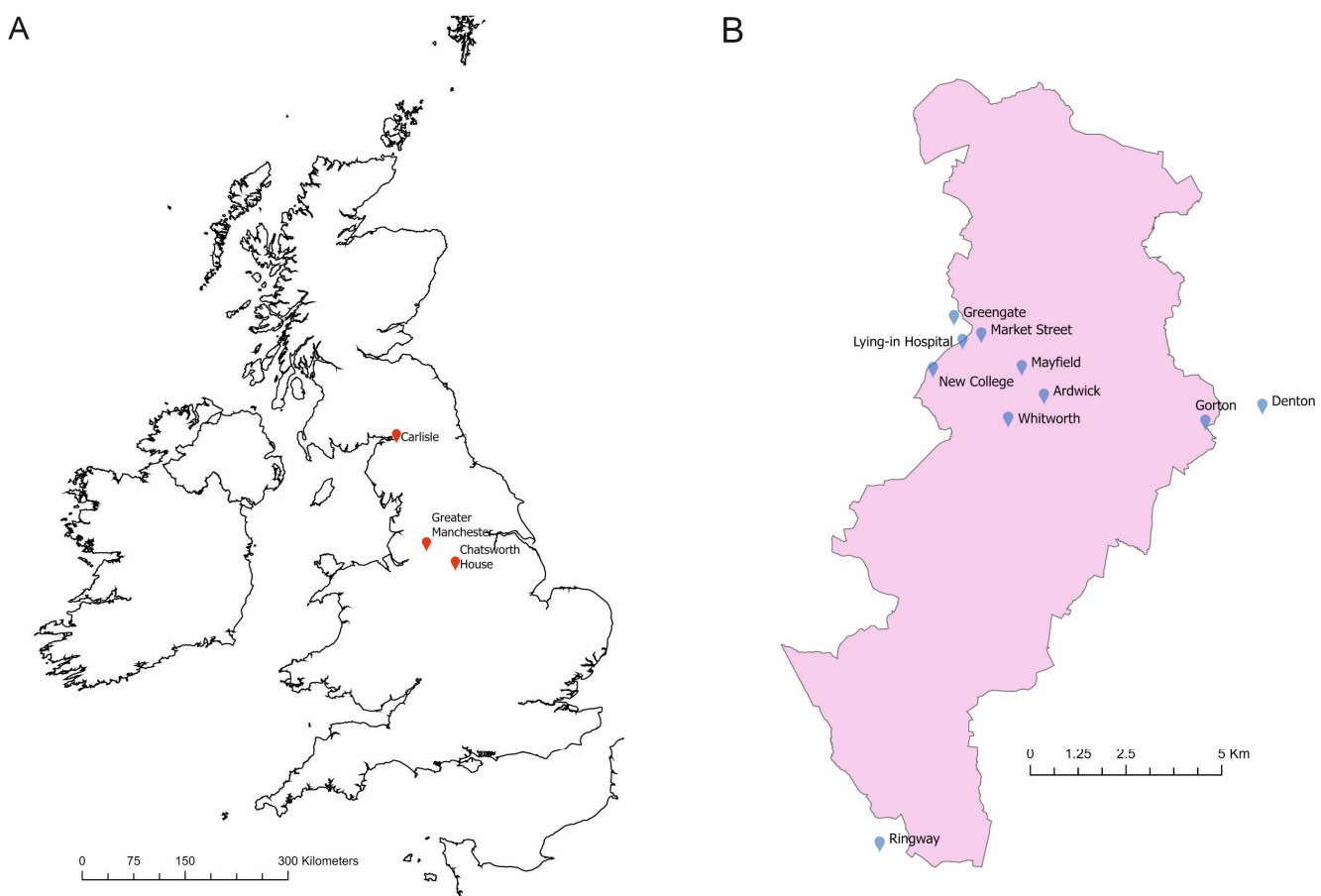

**Figure 1.** (**A**)—Locations of study sites (red points) and rainfall records used for homogeneity testing (blue dots); (**B**)—Location of meteorological stations used to reconstruct the composite Manchester rainfall record within the metropolitan borough of Manchester.

For the rainfall record to be deemed reliable enough to be used for an accurate analysis of long-term trends, it must be considered homogenous; in other words, any variations must be solely the product of natural climatic variations [32]. Non-natural causes of variation commonly afflict historic rainfall measurements, often arising from changes in gauge, location, observing practices, or even alterations in the surrounding environment. Inhomogeneities can sometimes be easily explained by reviewing the metadata, for example, a change in location; however, others are near impossible to explain, such as the growth of trees near to the gauge. To produce the most accurate series representative of the area, these inconsistencies must be addressed [21,32]. Several methods can be applied, dependent upon various factors; for example, the climatic variable being tested, the amount of data available, and the quality and quantity of the metadata. A number of homogeneity tests were applied as follows:

1.  A visual inspection of the plotted long rainfall series to identify inhomogeneities indicated by clear discontinuities (steps or breaks).
2.  A double-mass curve test [33] using the Manchester rainfall record and data from three series representative of sites in the region (Carlisle, Chatsworth House, and HadNWEP).

3. A standard normal homogeneity test (SNHT) [34] also using the Manchester, Carlisle, Chatsworth House, and HadNWEP rainfall records.
4. An analysis of station metadata to explore possible causes of inhomogeneities [18].

**Table 2.** Stages of linear regression. Details of correction factor (m), intercept (c), and strength of relationship between variables ($R^2$). Colour shaded by stage (X, A, B, C).

| Stage | Dependent Variable | Independent Variable | Overlap Period | Slope (m) | Intercept (c) | $R^2$ |
|---|---|---|---|---|---|---|
| X | Greengate, Salford | NA | NA | NA | NA | NA |
| | New College, Dawson St | NA | NA | NA | NA | NA |
| A1 | Mayfield | Lying-in Hospital | 1807–1811 | 0.998 | −0.101 | 0.893 |
| A2 | Mayfield | | 1830–1840 | 0.901 | 12.242 | 0.967 |
| B | Ardwick | Gorton Reservoir | 1860–1879 | 0.978 | 3.067 | 0.974 |
| C1 | Whitworth | Denton Reservoir | 1931–1957 | 0.932 | −2.539 | 0.979 |
| | | | 2003–2023 | | | |
| C2 | Denton Reservoir * | Ringway Airport | 1980–2003 | 0.973 | 1.267 | 0.964 |

* Adjusted Denton Reservoir values to primary station Whitworth.

## 4. History of Rainfall Recording in Manchester

Manchester enjoys a long and rich history of recording rainfall. Since the 18th century, there has been a continuous interest in keeping records, meaning that Manchester exists as one of the few places in the UK where one can draw upon over 200 years of data to observe the changing pattern of the weather through time [8]. The history begins with Dr. George Lloyd in 1765. Lloyd, an elected Fellow of the Royal Society and member of the Manchester Literary and Philosophical Society, recorded rainfall at Hulme (central Manchester) until June 1771 [8,35]. However, during data collection, only data up to the end of 1769 were retrieved from the UK Meteorological Office decadal rainfall books. There is then a 16-year gap in monthly rainfall totals; however, John Poole did record the number of days with rain, between 1769 and 1786, at Rhodes (near Middleton, approximately 10 km north of Manchester). Manley [8] incorporated Poole's observations into his record, remarking that the number of wet days were almost the same as the contemporary annual average. At this point it is necessary to note that Lloyd's curtailed record and Poole's wet day observations have not been included in the reconstruction. There does not exist a suitable neighbouring station to bridge the gap between the end of Lloyd's data and the next observer of monthly rainfall totals, and there is too much uncertainty regarding a conversion of wet days into monthly totals.

Therefore, this Manchester rainfall record commences with the measurements of George Walker, a cotton merchant from Greengate, Salford. Having started to record the number of wet days in 1783, Walker decided to set up a rain gauge in 1786. Manley [8] stated that Walker continued this practice until 1813 but observations post-1793 were not retrieved during data collection. It is at this point important to acknowledge the remarks of the next observer, the famous scientist Dr. John Dalton [36], who acquired Walker's records upon his death and concluded after inspection that he would prefer to keep his own record separate, for the following reasons:

> "On a comparison of our results for 8 subsequent years, I found his average exceed mine, by about 4 inches in the year... On inspecting his gage [sic], I had reason to think his mode of measuring the rain was not susceptible of sufficient accuracy, and suggest the same to him, with which he seemed to acquiesce."

This information is vital and may be used to explain, and later fix, any inhomogeneities that may be present in the early part of the record. The next set of measurements were taken by Dalton himself. Dalton is easily the most famous observer in the record; a chemist, physicist, and meteorologist, Dalton is known for his pioneering work in the field of atomic theory and research into colour blindness. An elected Fellow of the Royal Society and member of Manchester's Literary and Philosophical Society, Dalton moved to Manchester in 1793 to commence teachings of mathematics and natural philosophy at the 'New College',

a dissenting academy on Dawson Street (now Mosley Street). Having recorded observations of the weather in Kendal between 1788 and 1792, Dalton began to keep meteorological observations of temperature, rainfall, and pressure in Manchester from 1794 using a rain gauge located at the 'New College' [8]. These measurements contributed to a diary, which across 57 years, accumulated over 200,000 observations [37]. Dalton continued to record at this location until 1802, after which, the gauge was moved to Mayfield in the garden of a Thomas Hoyle, the father-in-law of Dalton's close acquaintance and fellow quaker: William Nield (later a Mayor of Manchester, 1841–1842). Hoyle was also a named subscriber to Dalton's [38] Meteorological Observations and Essays. Dalton ([36] p.15) published his Mayfield measurements as part of his "Observation on the Barometer, Thermometer, and Rain, at Manchester: From 1794 to 1818 inclusive" essay, describing the rain gauge and its location

> "The rain-gage [sic] has been all the time situate in the garden on the S.E. side of Manchester; it is twenty yards distant from any house or elevated object that can influence the fall of rain. The gage is a funnel of 10 inches diameter, and the top is surrounded by a perpendicular rim of 3 inches high, to prevent any loss by the spray; it is fixed in a proper frame with a bottle for the water, and it stands above 2 feet above ground."

(Dalton. 1819, p.495)

The record continued until 1840, by which time the ageing Dalton had been suffering from ill health for some years. Manley [8] suggested that Dalton's later measurements were perhaps less representative because of deficient catch caused by a block of terraced houses being erected on the west side of the garden in ca. 1836. However, comparisons with other simultaneous records at Hyde (J. Ashton), Market Street (J. Casartelli), and Ardwick Green (L. Buchan) do not reflect a marked decrease in catch at the site, and so the latter years of Dalton's record have been included in the reconstruction.

During the last decade of Dalton's record, Joseph Casartelli began keeping a rainfall record at 43 Market Street (1830–1852). Casartelli, an Italian immigrant and optician and philosophical instrument and hydrometer maker from the Lake Como region, later moved the gauge to Ardwick ([8]: p.94). According to Manley [8] this practice continued until 1888; however, data from 1880 onward were not retrieved during collection. An additional coeval 27-year record exists for this period, recorded by G.V. Vernon at Old Trafford; however, its usefulness is limited by the prevalence of missing data. Vernon [39] (p.199) admits as much himself in his submission to the proceedings of the Manchester Literary and Philosophical Society, stating that "the period 1850 to 1860 is very incomplete, owing to the month of August being deficient in this first six years of the period". Manley [8] describes Vernon's record as "rather uncertain" and based on these facts, the record was not considered for inclusion within the reconstruction.

J. Bateman, and latterly G.H. Hill, were recording rainfall at various sites across Manchester in the mid-1800s, including Gorton and Denton Reservoirs, on behalf of Manchester Waterworks Authority. These records are long and mostly continuous, with Denton still an active meteorological station today, but contain several issues limiting their usefulness in early parts of the record. Denton is less reliable for the first half of the record; between 1854 and 1928, the rain gauge was positioned at the level of an embankment and was consistently exposed to under-catch as a result of wind stress. Therefore, Denton's record is incorporated only into the composite series from 1929 onward, providing the most recent data since 1979–2023.

The Manchester University observatory at Whitworth Park started in 1893; data from the Whitworth Observatory were provided by the University to the Guardian newspaper for daily and weekly weather reports for the benefit of Manchester citizens, whilst also being forwarded to the Meteorological Office [40]. During this period, there were several different observers, including Captain Alan Morris Jones (c. 1919–1923), Arthur A. Jones (1924), H.E. Martin (1925–1933), A.E. Martin (1930–1936), and J. Hamilton (1934–1936). Toward

the end of its recording history, the Observatory was falling into disrepair having been subject to repeated and extensive vandalism in the park location; observations ceased in 1957, and the Observatory burned down the following year under mysterious circumstances [40]. The final station contributing to the composite record is Ringway Airport, which started in 1941 (although data were retrieved only from 1951) with the site discontinued in October 2004.

## 5. Station Selection and Analysis

### 5.1. Station Selection

Meteorological stations were chosen for the reconstruction of the Manchester rainfall record based on their longevity, continuity, and proximity to the primary (oldest) records for Manchester, following the method of Macdonald et al. [30]. The first year of the composite record is 1786, commencing with George Walker's observations in Salford, even though the earliest available monthly totals are those of George Lloyd from 1765. Lloyd's observations were not included in the reconstruction because they are succeeded by a 16-year gap before Walker's monthly records began, and there are no suitable data available from an overlapping neighbouring station to bridge the gap. Forming the basis of the reconstruction, three primary records were selected: Dr. John Dalton at Mayfield (1803–1840); Mr. Joseph Casartelli at Ardwick (1853–1879); and the University of Manchester at Whitworth Park Observatory (1893–1957). These records are at least 25 years in length, continuous, and located within proximity of one another (1.5 km). To complete the record, a further seven rainfall series were required, all located less than 15 km away from the primary stations (see Table 1). This provides the least number of different stations needed to construct a continuous rainfall record for Manchester. For the rainfall series used, only Mayfield (Mar–Apr 1807, Dec 1809, Jan–Feb 1811) and Denton Reservoir (1960–Jan 1979) contain periods of missing data, which were filled using linear regression analysis with overlapping records located within close proximity. The gaps in the Mayfield record were resolved using Dr. Thomas Henry Hanson's record kept less than 2 km away at Lying-in Hospital (now known as St. Mary's Hospital), and the periods of missing data within Denton Reservoir's record were infilled using the Ringway Airport series recorded 15 km SW from the site.

### 5.2. Statistical Comparison of the Primary Stations

Firstly, the three primary stations (Mayfield, Ardwick, and Whitworth) were compared in terms of their monthly mean with two standard errors calculated for a 27-year period (Figure 2). This period was chosen based on the length of the shortest record of the three: Ardwick (1853–1879). The first 27 years of the Mayfield (1803–1829) and Whitworth (1893–1919) series were selected. Figure 2 shows the variation in mean monthly rainfall for the three stations. There is generally good agreement, particularly in the first half of the year, with the greatest variation between stations observed in September and November. This is understandable when one considers the potential skew an extreme season/period may have on monthly means in such a short 27-year period, for example, the impact of the extremely wet Autumn of 1824 (second wettest on record) on the mean monthly November totals at Ardwick. However, in general, the two standard error bars reflect no significant difference in monthly rainfall totals at the three primary stations, justifying their inclusion in the composite rainfall record for Manchester.

### 5.3. Linear Regression Analysis

Nearby stations were identified to bridge the intervals between the three primary rainfall records based on the length of the overlap period, their reliability, and the strength of the relationship with the primary station. In every case, the correlation coefficient $R^2$ exceeded 0.8 (strong positive correlation) and in all but one case, the overlap period was at least 10 years in length (see Table 2). The linear regression analysis was performed in stages (A1–C2) using the primary stations as dependent variables (y) and neighbouring stations as independent variables (x) (using the greatest possible overlap) to infill gaps within

the primary records or to bridge gaps between them. Note that the first two series that contribute to the composite record, in stage X, did not undergo linear regression analysis because they do not overlap with the first primary station or any other suitable records for Manchester, and were therefore included without a correction factor being applied.

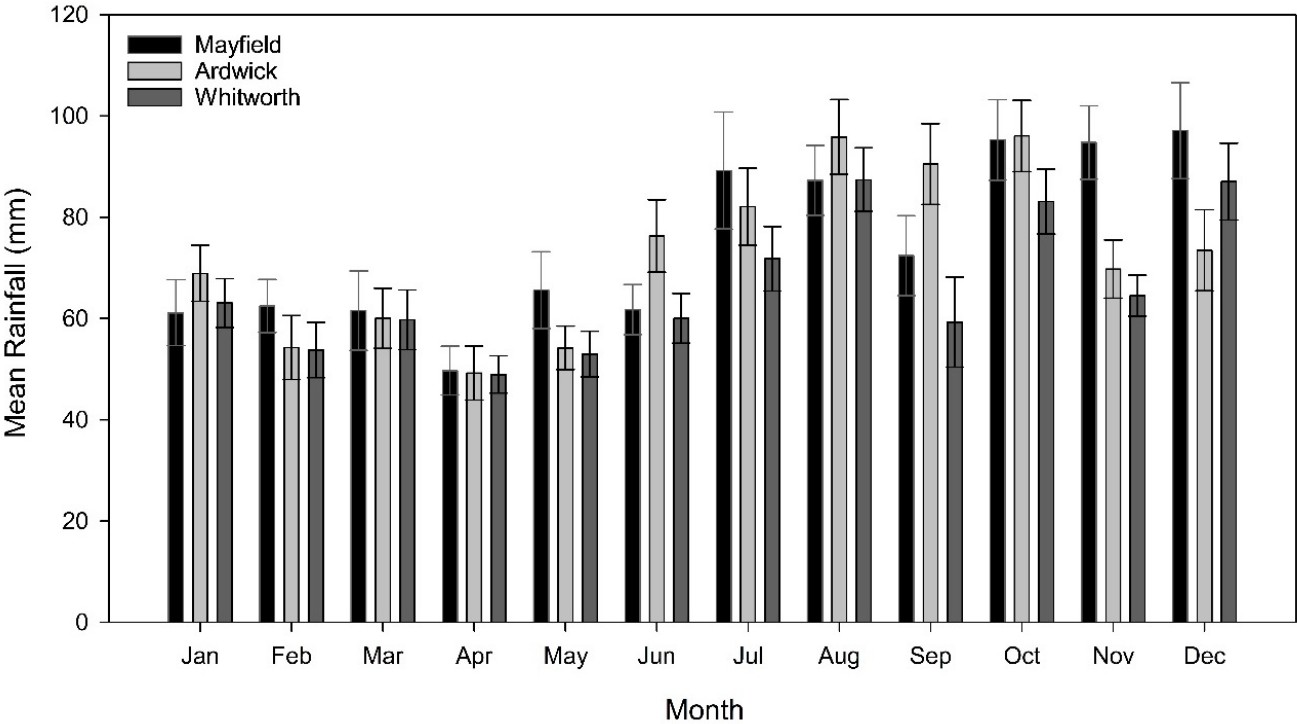

**Figure 2.** Comparison of monthly mean rainfall (Jan–Dec) over 27 years at the three primary meteorological stations (Mayfield, Ardwick, and Whitworth) with two standard errors shown.

Thus, the first stage of linear regression, A1, was undertaken to infill the months of missing data in Dalton's Mayfield record (Mar–Apr 1807, Dec 1809, Jan–Feb 1810). Fortunately, there was a suitable neighbouring station available with a period of overlap (1807–1811) with which to perform the linear regression analysis: Dr. Thomas Henry Hanson's record from Lying-in Hospital (now known as St Mary's). The regression equation was applied to adjust Hanson's data for the missing months and thus fill in the missing values at Mayfield. Next, the 12-year gap between the first two primary stations had to be bridged. To extend the Mayfield record, linear regression was performed with Casartelli's Market Street data for the overlapping period 1830–1840. The Market Street dataset was adjusted to achieve compatibility with the Mayfield record and to construct the bulk of the early homogenous record: 1803–1852. Then, Casartelli's Ardwick record required an extension to prolong it up to the beginning of the Whitworth record in 1893. In terms of the dependent variable, Bateman's record at Gorton Reservoir was chosen over his record at Denton Reservoir due to the issues related to wind exposure discussed previously. The final stage of homogenisation involved the extension of the Whitworth record up to 2022 using a combination of the Denton and Ringway records. Since the change in gauge location had alleviated the wind exposure issues at Denton and because it is a continuous record with a 37-year overlap period with Whitworth that continues to present day, it would have been an ideal station to complete the record. However, data for the period 1961–Jan 1979 are missing at Denton. So first, the linear regression analysis was performed between Whitworth (independent variable) and Denton (dependent variable) to adjust the Denton data for the periods 1958–1960 and from Feb 1979 to Dec 2022. Then, the adjusted Denton record served as the independent variable in a regression analysis with the Ringway record using the overlap period 1980–2003. Values from Ringway were adjusted to fill the gap in

the adjusted Denton record and therefore complete the continuous, homogenous rainfall record up to June 2023; however, only full years were used in the analysis.

### 5.4. Visual Identification and Correction of Inhomogeneities

A visual inspection of the plotted composite record exposed three periods of inhomogeneity, in sections X, A, and B, shown by sharp breaks in the 10-year moving average. Annual average rainfall totals were calculated for each stage, as well as annual percentage differences. The annual percentage differences of sections exhibiting inhomogeneity (X, A, and B) were compared to sections without visual inhomogeneity (C1 and C2). A comparison showed that the annual average difference for section X was 19.01%, section A was 12.78%, and section B was 3.51%, compared with sections C1 and C2. Rainfall values for sections X, A, and B were therefore decreased by the respective values to adjust for inhomogeneities in the record (Figure 3).

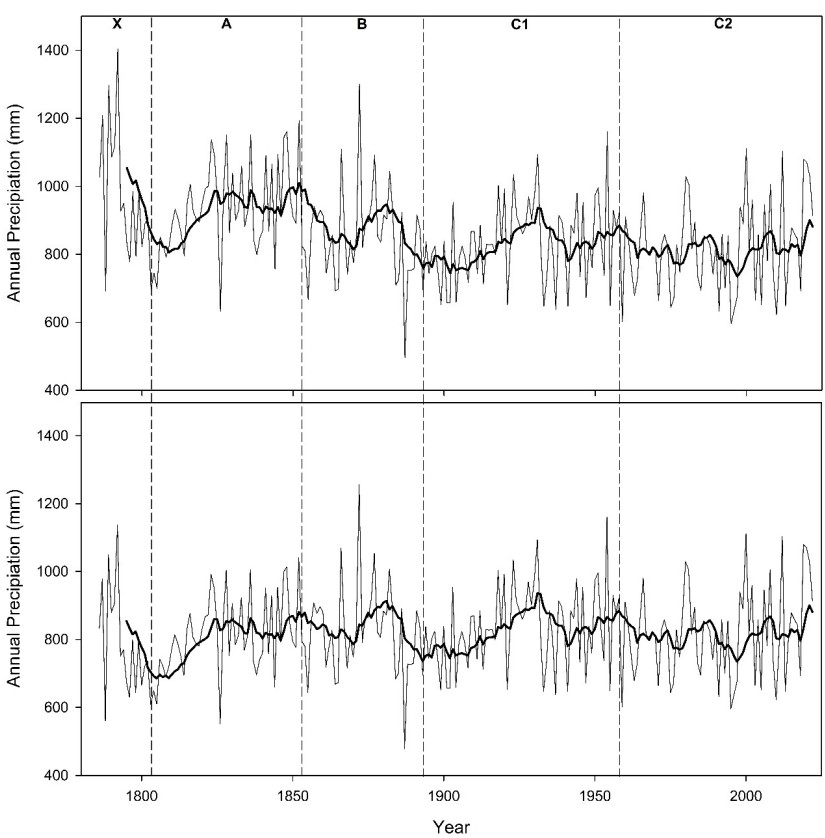

**Figure 3.** Composite rainfall record (1786–2022): raw recorded precipitation before (**top**) and after (**bottom**) adjustment factors of 19.01, 12.78, and 3.51% are applied to sections X, A, and B, respectively. The bold line is the 10-year running mean.

### 5.5. Homogeneity Testing

To assess the homogeneity of the constructed rainfall record for Manchester, a statistical cross-comparison with three other independent long-term series from northwest England was performed [3,18,33,34]. Records for Carlisle (1757–2012) [3], Chatsworth House (1777–2015) [9], and the HadNWEP region series (1873–2015) [10] were acquired. Table 3 indicates that Manchester is more comparable to Carlisle and Chatsworth House than HadNWEP, with regards to the total, mean, and standard deviation of rainfall for the common period 1873–2012. Generally, the values for the Manchester record are comparable to those of Carlisle and Chatsworth House and are consistently much lower than those of the northwest region (HadNWEP, Figure 4a–c). For the purposes of this study, the Carlisle, Chatsworth House, and HadNWEP records were selected for the next stage of homogeneity testing. The absence of independent long rainfall records can present challenges in

comparisons across series; Todd et al.'s [3] Carlisle record and Harvey-Fishenden et al.'s [9] Chatsworth House series were well suited as they have recently undergone revision and extension and are considered homogenous. To assess the strength and statistical significance of the relationship between the records, Pearson's product moment correlation was applied. The analysis revealed that the annual rainfall totals for Manchester with Carlisle (r = 0.52, *p* = <0.01), Chatsworth House (r = 0.62, *p* = <0.01), and the HadNWEP (r = 0.83, *p* = <0.01) all significantly correlate at the 95% significance level for the common period 1873–2012.

**Table 3.** Analysis of monthly rainfall variable for Manchester, Carlisle, Chatsworth House, and HadNWEP using the common period 1873–2012.

| Site | Total (mm) | Mean (mm) | Standard Deviation (mm) | Maximum (mm) | Minimum (mm) |
|---|---|---|---|---|---|
| Manchester | 114,885.0 | 68.4 | 35.4 | 216.7 | 1.4 |
| Carlisle | 111,965.8 | 66.7 | 35.5 | 242.3 | 2.0 |
| Chatsworth House | 118,936.3 | 70.8 | 39.8 | 284.4 | 1.0 |
| HadNWEP | 143,213.3 | 85.3 | 40.8 | 245.4 | 3.7 |

A double-mass curve test was performed; the test was used to assess the consistency of the rainfall records by comparing the test station (i.e., Manchester) with reference station(s) (i.e., Carlisle and Chatsworth House) [3,33] (Figure 4a–c). To examine relative homogeneity, cumulative precipitation from both stations were plotted; breaks in the slope of the curve indicate a change in the proportionality between the two datasets. Discontinuity may arise due to a variety of reasons, either from natural (e.g., meteorological) or from non-natural causes (e.g., changes in the gauge location, exposure, or observational practices). If changes are attributed to non-natural causes, then the rainfall record may be adjusted using coefficients determined from the double-mass curve [33]. Double-mass curves were plotted using the cumulative annual totals for Manchester, Carlisle, Chatsworth House, and HadNWEP using the longest possible overlap in each case (Figure 4j–l). The curves were visually inspected to identify changes in the gradient of the slope. The double-mass curves for Manchester and Carlisle (1786–2012, Figure 4j), Manchester–Chatsworth House and Manchester–HadNWEP (1873–2015, Figure 4j,k) show no obvious changes. Despite being a useful tool for testing relative homogeneity simply and efficiently, the double-mass curve test does not provide sufficient information regarding the timing of changes, the size of the shifts, or identification of which series caused the discontinuity. Furthermore, a key limitation of the technique is that it is vulnerable to human error and bias as the identification of a breakpoint "mainly depends on the judgement of the investigator" [41] (p.135).

Therefore, to provide a further analysis of the timing and magnitude of possible inhomogeneities in the record, Alexandersson's [34] SNHT was applied as in Todd et al.'s study [3]. A new standardised series of ratios (Qi) was generated for each of the three records using annual rainfall totals (Figure 4g–i) using Equation (1) from Alexandersson [34]):

$$Qi = (ui − ū)/su \tag{1}$$

where ū is the arithmetic mean value of the ratios (ui), and su is the sample standard deviation of this series. Therefore, the new series (Qi) has a zero mean value and unit standard deviation [34]. The test statistic (Tv) series was calculated using Equation (7) from Alexandersson [34], determined by the annual difference for two contiguous 25-year windows moved through the data series using the mean difference in precipitation between two stations [3]. The Qi series for Manchester and Carlisle depicts a rise during the first two decades of the 1800s, then a decline during 1880–1890s, followed by a rise in the mid-1900s (Figure 4d). Changes in the Qi series coincide with statistically significant breakpoints in the test statistic series (Tv > 7.75; n = 25; significance level 95%), based on the critical levels for the ratio test determined by Alexandersson (1986), at 1819 (Tv = 9.3), 1883 (Tv = 16.0), and 1908 (Tv = 15.0). The Qi series for Manchester and Chatsworth

House slowly increases to the early 1870s followed by a reduction during period 1870–1915, before a rapid rise and then gentle decline; the Tv results identify peaks in 1850 (Tv = 10.4), 1875 (Tv = 23.8), 1903 (Tv = 27.1), and 1943 (Tv = 22.7). The Qi series for Manchester and HadNWEP shows an increasing trend during the early 1900s, reaching a peak in the early 1930s. From the 1940s onward, there is an overall declining trend to the present, with a sharp reversal of this pattern observed in the early 1970s (Figure 4f). These trends again coincide with statistically significant breakpoints observed at 1908 (Tv = 12.0), 1933 (Tv = 12.6), 1948 (Tv = 12.4), 1973 (Tv = 10.3), and 1984 (Tv = 8.3).

An SNHT analysis for Carlisle and HadNWEP was performed (not presented graphically) to examine the relative homogeneity of the test series, and was required, given that the identified breakpoints involving Manchester yielded different outcomes. The Qi series exhibited a gradual decline between 1890 and 1910, a rise up to 1950, and then an overall decreasing trend to present, with a notable reversal and peak in the mid-1980s. Statistically significant breakpoints were observed at 1911 (Tv = 11.6) and 1985 (Tv = 12.7). The same analysis for Chatsworth House and HadNWEP identified 1902 (Tv = 8.8), 1915 (Tv = 11.3), and 1940 (Tv = 9.7).

*5.6. Identifying Causes of Statistical Inhomogeneity*

The early part of the record is composed of data from four different sources: Walker's Salford record, Dalton's earlier observations at the College on Dawson St, Dalton's later measurements at Mayfield, and to a lesser extent, data from Hanson's Lying-in Hospital record, which has been used to infill gaps at Mayfield; however these do not result in notable Tv scores, bar that in 1819 in comparison to Carlisle. It is likely that differences in instrumentation, site, and observing practices were the main causes of the sharp discontinuity in 1819 and 1850 [18], with two relocations in close temporal proximity to the 1850 peak (1841 and 1853). This assertion is supported by the remarks of Dalton [36] (p.18) when he concluded that Walker's Salford measurements were not made with "sufficient accuracy", but he does not elaborate as to why. Consequently, it is difficult to identify the exact source of inhomogeneity. Furthermore, this section contains a major drought event identified for England and Wales by Cole and Marsh [42], 1798–1808, which may have contributed to the observed inhomogeneity [3].

Later breakpoints were observed at 1883, 1908, 1911, 1933, 1943, 1948, 1973, 1976, and 1984/1985, although the latter breaks (1973, 1976, and 1984/1985) were identified only in comparisons involving HadNWEP and are likely to belong to this record. Analysing the potential causes of apparent inhomogeneity at 1883 and 1908–1911 requires examination of the meteorological stations contributing data to the Manchester series and their associated metadata. Firstly, the year 1883 falls in a gap in the primary station record between Ardwick and Whitworth. Data from Gorton Reservoir were used to infill the gap, selected based on the strong correlation (r = 0.97) with Ardwick (Table 2), and the fact that the other records available at this time, e.g., Denton, were affected by wind-related catch deficiency. The source of inhomogeneity may be the Gorton record; to test this, the gap was infilled using linear regression with the less reliable Denton record (pre-1929); however, the breakpoint was still observed. Further examination of the literature and the Manchester record identifies the 1880s as being particularly uncertain, with both Cole and Marsh [42], and Fowler and Kilsby [43], identifying the years 1883–1885 as droughts across northern England, whilst 1887 was the driest year on record in Manchester (Table 4). Todd et al. [3] observed a similar breakpoint at 1886 and emphasised the uncertainty surrounding the period. Multiple series identify breakpoints in the period 1880–1915, a period characterised by the notable extreme multi-decadal drought of 1890–1910 [3,44], which may help explain the high number of observed inhomogeneities within this period across multiple series. The period was described by Todd et al. [33, 3603] as being "the most sustained drought conditions in the instrumental record"; having identified similar breakpoints at 1886, 1909, and 1911, we noted that severe multi-decadal climate variations like this may explain observed breakpoints and inconsistencies.

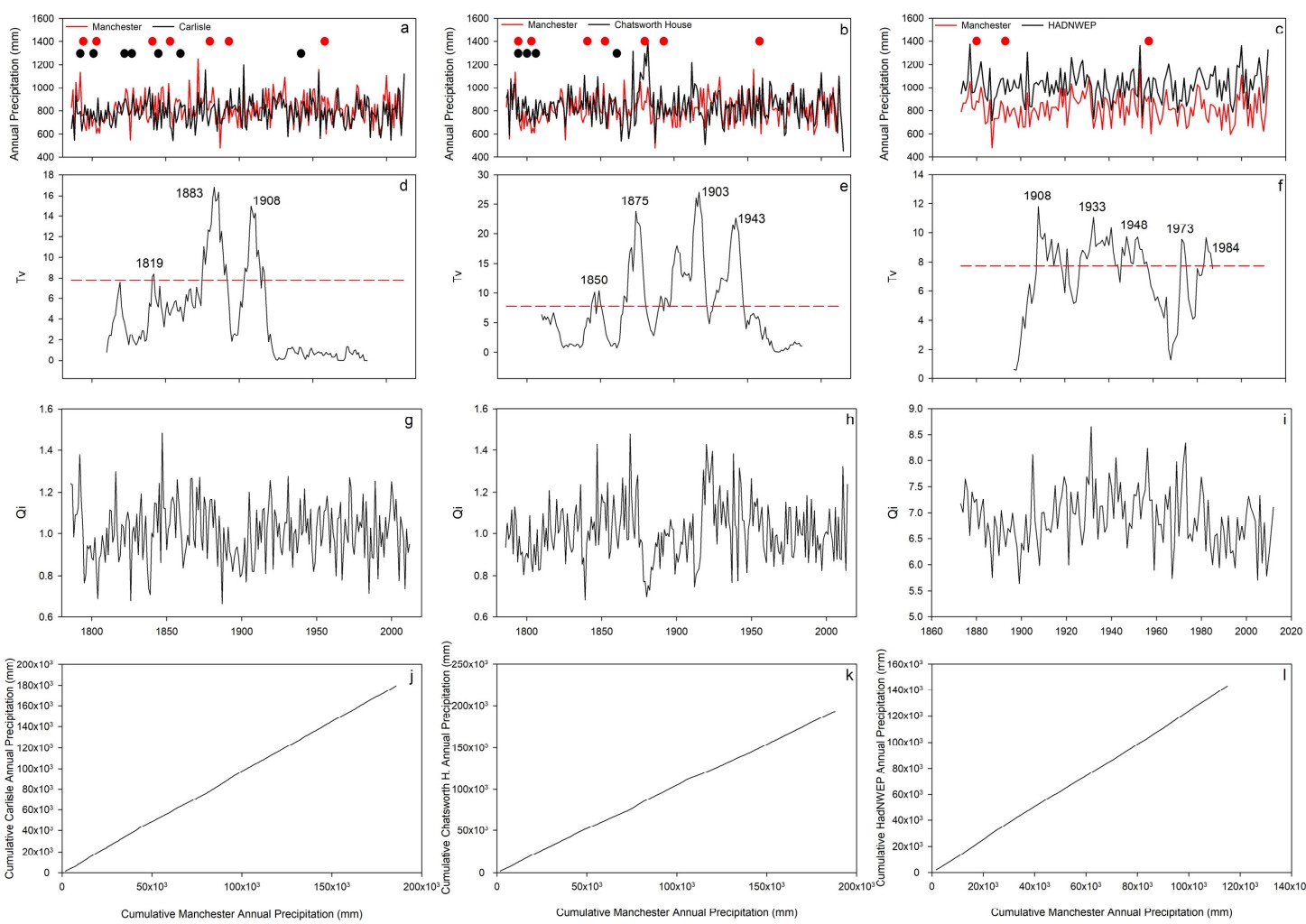

**Figure 4.** (**a–c**) Manchester precipitation series compared to other long regional series at Carlisle, Chatsworth House, and HADNWEP, respectively, with known station changes (coloured dots reflecting site); (**d–f**) Test statistic (Tv) series; the dashed line indicates significant scores (>7.75) with years of the peak score indicated; (**g–i**) ratio (Qi)) series (middle); and, double-mass curve of cumulative rainfall totals for (**j**) Manchester and Carlisle, 1786–2012, (**k**) Manchester and Chatsworth House, 1786–2015, and (**l**) Manchester and HadNWEP, 1873–2015.

**Table 4.** Top 10 ranked wettest and driest years and seasons during 1786–2022.

| Wettest Years | | Driest Years | |
|---|---|---|---|
| Year | Rainfall (mm) | Year | Rainfall (mm) |
| 1872 | 1255.5 | 1887 | 478.2 |
| 1954 | 1160.3 | 1826 | 552.1 |
| 1792 | 1136.5 | 1788 | 560.5 |
| 2000 | 1111.1 | 1995 | 595.6 |
| 2012 | 1102.3 | 1959 | 602.0 |
| 1931 | 1094.2 | 1803 | 608.6 |
| 2019 | 1079.2 | 1805 | 610.1 |
| 2020 | 1071.2 | 2010 | 622.8 |
| 1866 | 1069.3 | 1796 | 630.1 |
| 1877 | 1052.8 | 1991 | 631.9 |

The prolonged period of inhomogeneity in the Manchester and HadNWEP comparison (Figure 4b), marked by statistically significant breakpoints in 1933 and 1948, occurred during a 16-year period of the Whitworth record. Notable droughts were recorded within the UK in both 1933–35 [45,46] and 1946–49 [47]; however, the latter was poorly documented and reported on in the national press because of wartime reporting restrictions, as food rationing was in place. The station metadata provides no information clarifying the source of the detected inconsistency. Todd et al. [3] identified similar breaks during the 20th century and suggested that natural rainfall variability may be manifested as inhomogeneities in rainfall series from the region. This assertion is supported by the fact that that data for this period were sourced from a single station, therefore not subject to changes in location (Figure 4c), and the fact that higher levels of confidence are associated with the modern instrumental record [3]. It is notable though that the breaks identified with the HadNWEP series in 1973 and 1984 are not identified within either the Carlisle or Chatsworth House series; a break was identified in the Chatsworth House–HadNWEP analysis for 1982 and 1985, suggesting that these breaks may be a function of the HadNWEP series. It is also important to note that high natural rainfall variability exists between the reference (Manchester) and test stations (Carlisle and Chatsworth House) (see Table 3), and this may manifest itself as apparent statistical inhomogeneities in one of the two records [3]. Therefore, apart from a relatively short period of inconsistency, the Manchester record can be deemed generally homogenous.

## 6. Analysis of the Manchester Rainfall Record

The annual rainfall totals for the entire reconstructed Manchester rainfall record 1876–2015 are displayed in Figure 3. The overall mean annual rainfall was 820.8 mm, with a standard deviation of 124.9 mm. The ten wettest and driest years are listed in Table 4. The wettest individual year on record was 1872 with 1255.5 mm, and the driest was 1887 with 478.2 mm of rainfall, <50% of the annual average, and the driest calendar of the 19th century [42,48,49]). The drought had dramatic impacts across the whole of England, especially in the northwest, and in Manchester, the lack of water forced the closure of mills, quarries, and factories, putting thousands out of work. Summer 1800 was the driest season on record, a mere 41.6 mm of rain falling (more than 180 mm below the overall summer average). Hannaford [49] identified the first few years of the 19th century as being one of the top ten worst drought periods ever in the UK with the consequences felt nationwide. The Manchester record depicts several extreme seasons and years that contributed to this severe period: the winter of 1800, spring 1802, and the overall year of 1803 (the fifth driest total on record) (see Table 4). Other major extreme dry seasons exhibited in Table 4 include the droughts of summers 1976 and 1995, where rainfall deficiencies lasting more than 4–6 months accounted for agricultural losses of £180 million and >£500 million, respectively [42].

An examination of the entire record over time (Figure 3) shows an increasing trend in annual rainfall totals; however, the correlation coefficient with time at Manchester (r = 0.093) is weak and the trend is not significant at the 95% level (*p* = 0.2). A similar trend was also observed by Todd et al. [3] at Carlisle (r = 0.11), and for the wider northwest region (r = 0.084) in general (HadNWEP, [10]), whilst the signal at Chatsworth House is one of little change (r = −0.007).

The seasonal rainfall totals identify that summer (27.6%) and autumn (28.6%) exhibit close agreement and were the wettest, followed by winter (24.5%) and spring being the driest (19.3%). This overall trend was similarly observed at Carlisle by Todd et al. [3] and at Durham (1850–2012) [28] and suggests that rainfall seasonality has remained relatively constant over at least 150 years across the north of England. The seasonal statistical analysis (linear regression, correlation, and significance testing at the 95% level) undertaken for the entire Manchester record indicates that winter is the only season exhibiting a significant positive trend (r = 0.17, *p* = <0.01). The correlation coefficient for spring is also weakly positive (r = 0.12) but insignificant (*p* = 0.07), whilst both summer (r = −0.07, *p* = 0.3) and autumn (r = −0.02, *p* = 0.77) display a slightly negative, albeit insignificant, trend over time. Variations in decadal scale seasonal rainfall totals are shown in Figure 5. The seasonal totals on the decadal timescale are variable, with autumn values fluctuating by as much as 100 mm within 20 years in the early 20th century. At the beginning of the record, autumn is the wettest season, and for the remainder of the late 18th century, autumn continues to be the wettest season except for a brief period in the late 1790s when summer overtakes it. During this period, spring is generally the driest month, at times by more than 50 mm on average. Throughout the 19th century, the wettest season continued to fluctuate between autumn and summer, with autumn being marginally the wettest overall. This seasonal pattern is comparative to that of Carlisle [3], with spring generally the driest and summer and autumn generally the wettest.

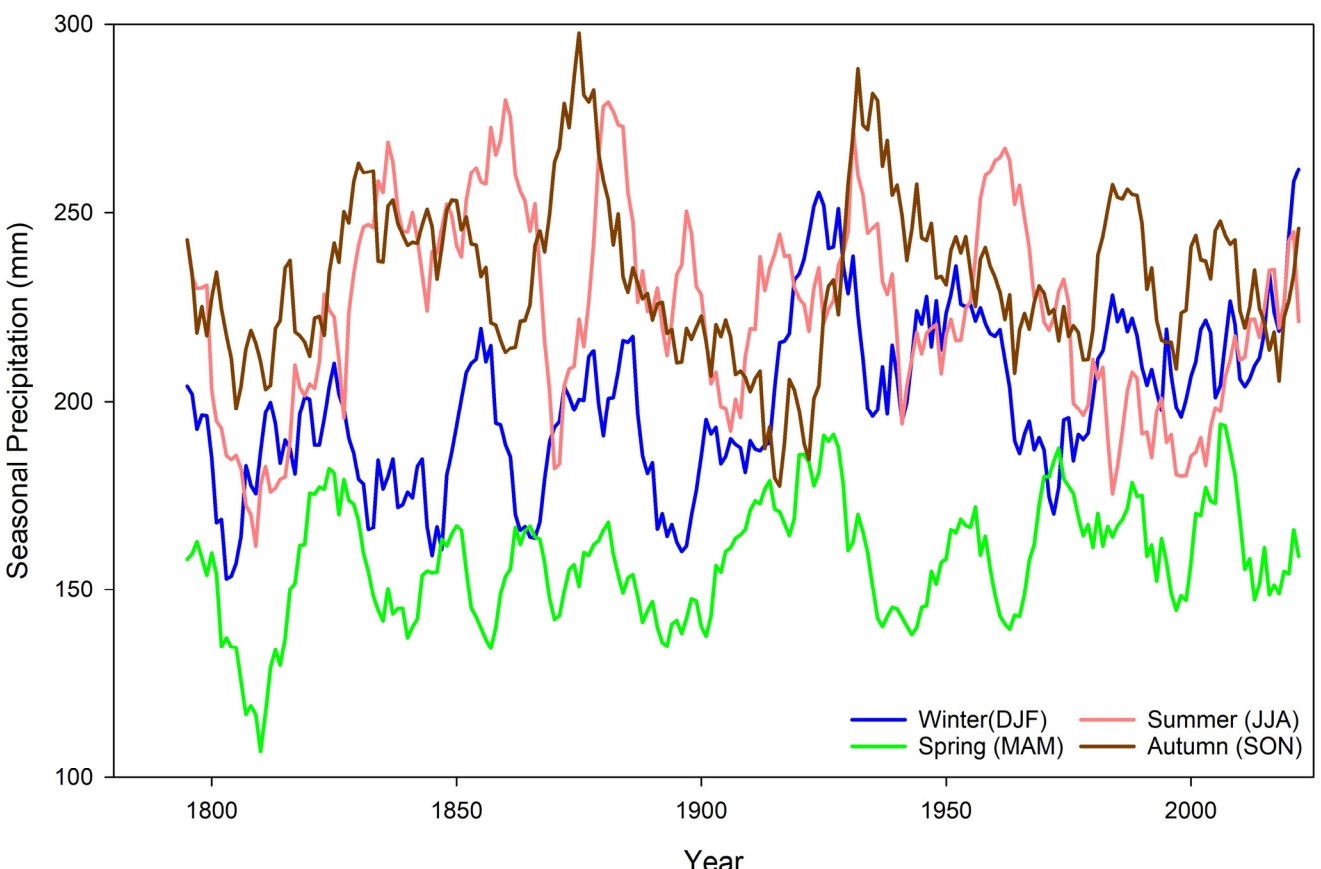

**Figure 5.** Seasonal rainfall totals at Manchester as 10-year moving averages.

The complete record illustrates several notable extreme wet and dry periods, and a number of well-known flood and drought events are listed in the top-ten ranked list of seasonal and annual extremes (Table 4). The wettest year on record was 1872 with 1244.4 mm, more than 400 mm higher than the average annual total. Known as the 'Great Flood', the particularly wet summer season (395.5 mm) was characterized by severe flooding that caused extensive damage in Manchester. During the wettest month of July, the River Medlock (a tributary of the Irwell) broke its banks after two days of torrential rain. The event was so dramatic that it was also memorialised through a local folksong "The Great Flood", with the local newspaper, The Manchester Courier (15 July 1872) [50] describing the devastation as follows:

> "It was about half past twelve when the floods came... the banks of the Medlock were overflowed to such an alarming extent and the first intimation of the flood was the sweeping away of a footbridge near to Philips Park... The flood increased in depth and power, and at a length swept in a fierce torrent over a large portion of ground apportioned to the Roman Catholics at the Bradford Cemetery carrying away not only tombstones but actually washing out of their graves, a large number of dead bodies. Indeed, from the first indication of danger, so far as works on the banks of the Medlock were concerned, dead bodies were observed floating down the river, and those watching could easily see that the bodies had been disinterred out of the Bradford cemetery. It is impossible to calculate how many had been swept out of their final resting place but the number is not short of fifty."

Table 4 shows that four of the top ten wettest years on record have occurred since the turn of the century, 2000, 2012, 2019, and 2020, but also that two of the top-ten wettest years occurred during the 1870s: 1872 and 1877. The top ten driest years shows that one of the driest occurred this decade in 2010, but the only decade with two years in the top ten is the 1800s (1803 and 1805), which relates to notable major droughts and prolonged drought periods [44,49]. The wettest season on record was autumn 2000 with >420 mm of rain, more than 200% greater than the seasonal long-term average (Table 4). Since records began in 1766 for England and Wales, the wettest autumn period was characterised by UK-wide flooding. In Manchester, there were only three days of negligible rainfall over the whole of October and November. Other notable events depicted in the ten wettest years include 1954, which saw six hundred homes flooded in Salford, 1866 ('the year of the great flood') where rain fell for three continuous days, and the terrific storms of winter 1877 when the river Irwell rose 8–9 ft above the normal level, flooding the surrounding towns and low-lying regions [51,52].

## 7. Summary

The UK enjoys a great history of observing and recording the weather, and in Manchester, the practice dates back to the mid-18th century. A composite Manchester rainfall record, incorporating even the earliest observations, was originally developed by the climatologist Gordon Manley in the 1970s; however, there exist many issues surrounding the quality of the data, thus lowering confidence in the original record. However, given the quality and quantity of available data for Manchester, it is possible to reassess and extend the record to produce a homogenous composite rainfall record for 1786–2022. Here, the 235-year record produced for Manchester was fully digitalised, and currently ranks as the second-longest series available for northern England after Carlisle (1757–2012) [3].

The data were collected from various sources, from archival decadal rainfall books available at the UK Meteorological Office archives in Exeter to the online data repository of the BADC. Once the data were transcribed and digitalised, to form the basis of the reconstruction, primary stations were selected regarding their longevity, continuity, and proximity to one another. To infill and bridge gaps in the primary record, meteorological stations were selected based on their location and data quality. Linear regression analyses were applied to the data to infill and bridge the gaps in the primary station records and

to ultimately construct a composite series for Manchester. To ensure comparability of early observations with contemporary stations, homogenisation techniques were applied. For example, a visual analysis of the plotted complete record identified three periods of inhomogeneity (X, A, and B), and correction factors were applied based on the annual percentage difference to parts of the record considered visually homogenous (C (1–2)). After adjustment, several homogeneity tests were utilised to ensure that the record is consistent and indicative of the natural climatic variations in northwest England.

The relative homogeneity of the composite series was tested through cross-comparisons with two other series from the northwest region. This is a useful technique, but the high natural rainfall variability of northwest England increases the difficulty of drawing conclusions. The Manchester record was deemed to be homogenous overall. However, two periods of inconsistency were identified between the Manchester series and Carlisle and HadNWEP records following the application of the SNHT approach [34] and the double-mass curve test [33]; 1786–1819, and 1883–1911. Although no specific cause could be identified during the metadata examination, the four location changes and the doubts raised by Dalton [36] over Walker's record at Salford are the most probable sources of the uncertainty observed in the earlier part of the record [18]. The later period, 1883–1911, was also subject to a change in site, from Gorton Reservoir to Whitworth Park, but there is a lack of evidence in the available metadata to clarify the cause of the uncertainty. Todd et al. [3] identified a similar period of inconsistency (1886–1911) during the homogeneity testing of the Carlisle record and emphasised the potential impact of the multi-decadal extreme drought 1890–1910 in the region. Later breaks identified in comparisons with HadNWEP for the 20th century were attributed to the natural rainfall variability of the region, which may be detected as statistical inhomogeneities between the rainfall records.

The analysis of the overall rainfall record indicated that there was a weak increasing trend in annual rainfall totals over time; however, this trend was shown to be insignificant. The only significant trend identified was that of an increase in winter rainfall, with no significant trends exhibited for the other seasons. This scenario was similarly observed by Todd et al. [3] at Carlisle, and the increasing winter rainfall pattern is in agreement with trends depicted in other long rainfall records for the UK, increasing confidence in the reconstruction (e.g., England and Wales [53]; western UK [54]; Durham Observatory [28]). The presence of this pattern in series from across the UK indicates that the shift to wetter winters is a widespread, national-scale change in rainfall patterns. A number of these records also observed a simultaneous decreasing trend in summer rainfall; despite a weak decreasing trend being observed at Manchester, the general trend was found to be statistically insignificant.

The 235-year record captures a number of well-documented extreme wet (e.g., 'The Great Flood' in summer 1872, and autumn 2000) and dry (e.g., the summers of 1887 and 1800) seasons/years, identified in the literature as being particularly severe [44,49,51,55]. The variable nature of extreme seasons, which accounts for an even distribution of extreme years throughout the record whilst also seeing three of the driest seasons ever recorded occurring in just the first seven years of the 19th century (spring 1806, summer 1800, and autumn 1805), is testament to the long-term natural variability of climate. Studying the record can serve to place recent extremes, in terms of both timing and severity, within a wider context; given that four of the wettest years have occurred since 2000, this offers important insight. Considering the impact of extreme wet and dry seasons on society, for example the cost of flooding and drought, a better understanding of the return periods of such events is important for informing the management of flood risk and water resources [27,29]. As mentioned previously, given that there are few long records available for the northwest, and indeed the UK in general, that predate the 19th century, this series is extremely valuable. Individual records, such as the one constructed, can offer valuable insights; however, there is also potential for incorporation into larger networks, which can provide a better understanding of regional- and even national-scale climate variation [5,56].

Finally, there is potential for this homogenous record to be used to test the homogeneity of other regional records in the future.

**Author Contributions:** All authors contributed to all stages of the research and production of the manuscript (data collection, analysis, and writing). N.M. conceptualised the initial research. All authors have read and agreed to the published version of the manuscript.

**Funding:** No specific funding supported this research.

**Data Availability Statement:** The rainfall record for Denton was acquired from the Environment Agency. The historic archive materials are housed in archives as noted within the text. The HadNWDP series is available from the Met Office (https://metoffice.gov.uk/hadobs/hadukp/data/ranked_monthly/HadNWEP_ranked_monthly.txt, accessed on 9 January 2024). The composite datasets presented in this article are part of an ongoing study. Requests to access the datasets should be directed to the lead author.

**Conflicts of Interest:** The authors declare no conflicts of interest. The funding sponsors had no role in the design of the study; in the collection, analyses, or interpretation of data; in the writing of the manuscript, and in the decision to publish the results.

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
