# Peer review of "Reassessing and Extending the Composite Rainfall Record of Manchester, Northwest England: 1786–Present"

_climate, doi:10.3390/cli12020021_

Round 1
Reviewer 1 Report
Comments and Suggestions for Authors
No comments and suggestions.
Author Response
Thank you for the positive feedback
Reviewer 2 Report
Comments and Suggestions for Authors
The paper presents a monthly composite rainfall record for the area of Manchester, northwest England. Using information from multiple sources, a continuous series of monthly rainfall data is created, ranging from 1786 to present. The paper contains many historical details about extreme rainfall events and severe droughts. It is well written and easy to follow. It uses mainly visual methods without entering deep into pure statistical-mathematical methods. Minor revisions required for publishing.
Line 70, 80, 126, 129 etc : Harvey-Fishenden…. Not in the reference list. The same reference is repeated in many places.
Line 102: Folland…. Not in the reference list
Line 109: Murphy…. Not in the reference list
Line 125: Burt… Not in the reference list
Live 179: “Colour shaded…” Probably belongs to Table 2 title.
Line 186, 321: Macdonald…. Not in the reference list
Line 476: “18070” Typo?
Line 478: Labels in figure 4l are on different years.
Line 520: Fowler……. Not in the reference list
Line 535: All three references in the line are not in the reference list
Line 597, Figure 5: A stacked column diagram could give clearer information: if possible, change the diagram type.
Line 749: Remove number from line and rearrange reference numbers.
Line 759: Remove number from line and rearrange reference numbers.
Author Response
Many thanks for the comments, these have been addressed:
I have added the requested references for: Harvey-Fishenden; Folland; Fowler; Murphy; Burt; Dayrell; Barker; Lennard; & Macdonald - these appear to have dropped out in the update to Mendeley cite. Apologies.
179: “Colour shaded…” Probably belongs to Table 2 title. Corrected - moved to Table 2.
476: “18070” Typo? Corrected 1870s
478: Labels in figure 4l are on different years. - Corrected in text
597, Figure 5: A stacked column diagram could give clearer information: if possible, change the diagram type. - Thank you for the suggestion, having tried this I felt it was less helpful as the changing patterns through time were less evident.
749: Remove number from line and rearrange reference numbers. - Formatting corrected
759: Remove number from line and rearrange reference numbers. - Formatting corrected